# Evaluating variant pathogenicity prediction tools to establish African inclusive guidelines for germline genetic testing

Kangping Zhou [1], Kazzem Gheybi [1], Pamela X. Y. Soh [1] & Vanessa M. Hayes [1,2,3] ✉

## Abstract

**Background** Genetic germline testing is restricted for African patients. Lack of ancestrally relevant genomic data perpetuated by African diversity has resulted in European-biased curated clinical variant databases and pathogenic prediction guidelines. While numerous variant pathogenicity prediction tools (VPPTs) exist, their performance has yet to be established within the context of African diversity.

**Methods** To address this limitation, we assessed 54 VPPTs for predictive performance (sensitivity, specificity, false positive and negative rates) across 145,291 known pathogenic or benign variants derived from 50 Southern African and 50 European men matched for advanced prostate cancer. Prioritising VPPTs for optimal ancestral performance, we screened 5.3 million variants of unknown significance for predicted functional and oncogenic potential.

**Results** We observe a 2.1- and 4.1-fold increase in the number of known and predicted rare pathogenic or benign variants, respectively, against a 1.6-fold decrease in the number of available interrogated variants in our European over African data. Although sensitivity was significantly lower for our African data overall (0.66 $vs$ 0.71, $p = 9.86E\text{-}06$), MetaSVM, CADD, Eigen-raw, BayesDel-noAF, phyloP100way-vertebrate and MVP outperformed irrespective of ancestry. Conversely, MutationTaster, DANN, LRT and GERP-RS were African-specific top performers, while MutationAssessor, PROVEAN, LIST-S2 and REVEL are European-specific. Using these pathogenic prediction workflows, we narrow the ancestral gap for potentially deleterious and oncogenic variant prediction in favour of our African data by 1.15- and 1.1-fold, respectively.

**Conclusion** Although VPPT sensitivity favours European data, our findings provide guidelines for VPPT selection to maximise rare pathogenic variant prediction for African disease studies.

## Plain language summary

Genetic testing for inherited diseases often excludes African populations due to a lack of relevant data, leading to European-biased guidelines. We address this issue by evaluating 54 variant pathogenicity prediction tools (VPPTs) using genetic data from 100 men with advanced prostate cancer, split evenly between Southern African and European ancestries. The aim is to identify which tools work best for African genetic data. While some tools performed well regardless of ancestry, others were more effective for specific populations. The study highlights the need for African-specific guidelines to improve disease prediction and health equity. The findings can help tailor genetic testing tools to better serve African populations, potentially leading to more accurate disease predictions and better health outcomes in the future.

While gene-specific and gene panel genetic testing for rare and common diseases is readily accessible to most economically stable countries, this is not the case for Africa. As such, genetic screening for public health intervention is lacking across the continent. A literature review of cancer genetic screening within Sub-Saharan Africa revealed that only 15 (31.25%) of 48 countries were represented in published genetic studies, with the vast majority limited to known breast cancer genes *BRCA1* and *BRCA2*[1]. Even within ethnically diverse countries, genetic testing for the most common diseases is tailored for people of European ancestry[2], including for prostate cancer (PCa), the disease example used in this study. Specifically, men of

[1]Ancestry and Health Genomics Laboratory, Charles Perkins Centre, School of Medical Sciences, Faculty of Medicine and Health, University of Sydney, Camperdown, Sydney, NSW, Australia. [2]Manchester Cancer Research Centre, University of Manchester, Manchester, UK. [3]School of Health Systems and Public Health, Faculty of Health Sciences, University of Pretoria, Pretoria, South Africa. ✉e-mail: vanessa.hayes@sydney.edu.au

**Table 1 | Demographic and clinicopathological characteristics and benchmark (positive and negative) datasets by genetic ancestry**

| Total (*n*) | | African 50 | European 50 |
|---|---|---|---|
| Country of origin (*n*) | | South Africa (50) | Australia (49), South Africa (1) |
| Genetic ancestral fraction % (*n*) | | >99% (50) | >99% (47), 97% (1), 90% (1), 74% (1) |
| Mean age (range) years | | 69.0 (53–99) | 62.4 (50–72) |
| Mean PSA (range) ng/mL | | 448.3 (8.21–4841) | 12.8 (3.5–31.8) |
| ISUP grade (*n*) | 3 | 0 | 1 |
| | 4 | 12 | 10 |
| | 5 | 38 | 39 |
| Positive dataset (*n*) | ClinVar | 39 | 99 |
| | InterVar[a] | 135 | 146 |
| | Merged | 158 | 202 |
| Negative dataset (*n*) | ClinVar | 15,272 | 21,356 |
| | InterVar[a] | 32,922 | 93,003 |
| | Merged | 41,045 | 103,886 |

[a]Using ACMG-AMP guidelines.

African ancestry are at the greatest risk for disease presentation at a younger age with more aggressive pathology, including associated mortality compared with men of non-African descent[3,4]. Yet, there is currently no consensus on PCa germline testing for Africans[5]. The reason, as with many other diseases, is a lack of sufficient data[6]. Addressing the health equity gap for PCa, thereby contributing to improve the region's health metrics[7], we urge for caution in viewing Africa and Africans in singular terms[8], as evidenced by over 2,000 representative language groups and the continent at the epicentre of human genetic diversity[9,10].

In 2015, the American College of Medical Genetics and Genomics (ACMG) and the Association for Molecular Pathology (AMP) proposed a panel of 28 assessment criteria to categorise genetic variants as pathogenic, likely pathogenic, uncertain significance, likely benign or benign[11]. These criteria are based on population, computational/predictive, functional, and segregation data, and include data from databases (i.e. sequence database, population database, variant/disease-specific database), as well as patient information such as family history. However, the ACMG-AMP guidelines come with their own set of reproducibility challenges ranging from the gathering of information, criteria of interpretation, algorithm and cut-offs used for database preferences. Addressing these challenges, the Clinical Interpretation of Genetic Variants or InterVar tool was created, which, based on 18 of the ACMG-AMP criteria, is aimed at minimising human error and, as such, enables clinical interpretation[12]. In contrast, the National Centre for Biotechnology Information variant database ClinVar[13], relies on tested and validated functional effects of variants from external submitters for the determination of pathogenicity, providing classification as [likely] pathogenic, [likely] benign or uncertain significance[14]. However, both the ACMG-AMP guideline and database-classified pathogenicity prediction approaches are constrained by the limited availability of African-derived genomic resources and associated variant classification, questioning the suitability of established prediction criteria and hampering database inclusion[15]. This is evidenced most recently when interrogating for 20 genes contributing to current PCa germline testing panels[16]. Compared with non-Africans, we found Southern African men at 3-fold greater risk for aggressive disease presentation[17], to be 2-fold less likely to present with a known pathogenic variant (PV), with an elevated number of coding variants of unknown significance (VUS)[16]. Given that experimental interpretation for variant disease pathogenicity is tedious and time-consuming, it is critical to evaluate the applicability of in silico variant pathogenicity prediction tools (VPPTs) for evaluating the functionality of African-relevant VUS.

Here, having access to published African-*vs*-European ancestral whole genome sequenced variant data generated and called using a single technical and analytical pipeline for men presenting with National Comprehensive Cancer Network (NCCN) criteria for PCa germline testing[18], we test 54 VPPT algorithms[19,20]. Focusing on pathologically matched and genetically non-admixed patients, balanced for African (*n* = 50), specifically Southern African, and European ancestries (*n* = 50), largely Australian, we provide a thorough evaluation of VPPT performance for all whole genome germline rare variants (minor allele frequency [MAF] < 1%), make ancestry-specific VPPT-derived workflow recommendations, with testing showing an increase in VUS pathogenic prediction for African patients.

## Methods

### Inclusion and ethics statement

The data presented in this study have been made available by the Southern African Prostate Cancer Study (SAPCS) Data Access Committee and with study approval granted by the Garvan/St Vincent's Prostate Cancer Biobank. Patients were recruited adhering to the principles of the Helsinki Declaration, providing informed consent as stipulated by approvals granted from the University of Pretoria Faculty of Health Sciences Research Ethics Committee (HREC) in South Africa (including US Federal wide assurance FWA00002567 and IRB00002235 IORG0001762; #43/2010) and St Vincent's HREC in Australia (#SVH/12/231), with approval for genomic interrogation provided by the St. Vincent's HREC (#SVH/15/227) and Human Research Protection Office of the US Army Medical Research and Development Command E02371 (TARGET Africa) and E03280 (HEROIC PCaPH Africa1K).

### Germline variant data source and patient inclusion criteria

Whole genome variant data was derived from deep sequenced (average 46X coverage, range: 30-97x) whole germline genomes from an African-*vs*-European ancestral aggressive presenting PCa resource made accessible via the European Genome-Phenome Archive (EGA; https://ega-archive.org) and including men of Southern African (EGAD00001009067) and European ancestry (EGAD00001009066)[18]. Noting NCCN guidelines, which advises for germline genetic testing for men with metastatic, recurrent or high-risk localised PCa, defined as Gleason score derived International Society of Urological Pathology (ISUP) group grading 4 and 5, and/or prostate-specific antigen (PSA) > 20 ng/mL, regardless of family history[21], patient selection was biased towards ISUP defined advanced disease presentation and balanced for ethnic representation (Table 1). Representing the Southern African population identifier, our study included 50 Black South Africans (100% ISUP 4–5), while presenting the European identifier, our study included 49 Australians and 1 White South African (98% ISUP 4–5, with a single European man presenting with ISUP 3). Notably, PSA is excluded as an inclusion criterion as a consequence of the elevated overall PSA levels at presentation in our African over European men (mean 448.3, 22-fold greater than the NCCN European-based guidelines for aggressive disease, versus 12.8 ng/mL, *p* = 0.001). Elevated PSA levels concur with previous reports for the SAPCS for men with and without PCa[3,17]. Additionally, Black South African men present at diagnosis over 6 years later than our European Australian patient presentation at surgery, which has largely been attributed to regional differences in PSA screening practices. All sequenced data were generated and processed using a single technical and analytical pipeline, allowing for direct comparative analyses, while patient ancestries were verified using 7,472,833 markers across the genome and fastSTRUCTURE v1.0 population substructure analysis[22], allowing for selection of patients representing exclusively African or European genetic ancestries.

### Benchmark positive and negative datasets for VPPT testing

Single-nucleotide variants (SNVs) were extracted for our 50 African (*n* = 19,045,878) and 50 European ancestral advanced PCa patients

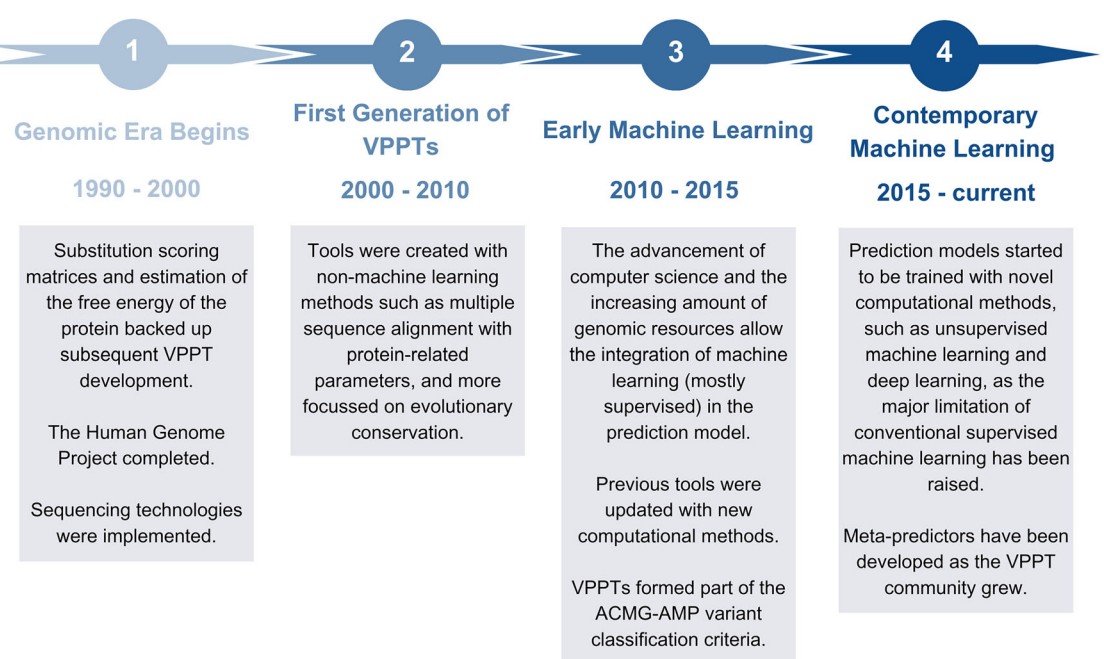

**Fig. 1 | 35-year timeline depicting major historical events in the development of in silico variant pathogenicity prediction tools (VPPTs).** Major historical events by timelines shaping VPPT development and defined as 1990-2000 (Genomic Era Begins), 2000–2010 (First Generation of VPPTs), 2010–2015 (Early Machine Learning) and 2015–current (Contemporary Machine Learning).

($n = 11,811,487$), representing 4,465,388 and 3,752,976 rare variants (MAF < 1%), respectively. Rare variants were further annotated using ANNOVAR[23] and dbNSFP v4.7[24], to establish ClinVar classifications (version: 20240611)[13], while InterVar (version: 2021-08, https://github.com/WGLab/InterVar)[12] was run to establish the NCCN guideline pathogenic predictions. For each ethnicity, positive and negative datasets were generated using known variant ClinVar classifications, defined as Pathogenic or Likely pathogenic (PVs) and Benign or Likely benign (benign variants, BVs), respectively, and as defined using InterVar ACMG-AMP guideline-driven prediction (Table 1). After removing overlapping variants, 158 African and 202 European-derived PVs were represented in the positive and 41,045 and 103,886 BVs in the negative merged dataset, respectively.

**Predicting VPPT performance criteria**

The 54 VPPTs, broadly categorised as (i) multiple sequence alignment (MSA), the earliest prediction method developed at the completion of the Human Genome Project and used to evaluate (interspecific) evolutionary conservation; (ii) protein structural/functional parameters, which evaluate how variants affect proteins' physical structure and in turn their functions; (iii) supervised machine learning (ML), introduced in the early 2010s, uses positively and negatively labelled model training datasets for variant prediction; (iv) unsupervised ML, which does not rely on labelled database; (v) deep learning (DL), a subset of ML that is powered by artificial intelligence to automate feature extraction and as such requires a larger training dataset and more complex computation; and lastly (vi) meta-predictors, that integrate features and prediction scores from multiple VPPTs, often combined with their in-house ML methods (Fig. 1), were included for prediction performance assessment and further defined by their year of development and method of prediction (Table 2). Prediction scores and results were similarly derived from dbNSFP v4.7 via ANNOVAR[23]. If available, the directly predicted classification of variant pathogenicity was used (e.g. T for Tolerant, D for Deleterious), while default/recommended numeric cut-off scores were applied to other tools.

The predictive power of 54 VPPTs for African and European variant data was assessed by the standard performance metrics including sensitivity (true positive), specificity (true negative), false positive rate (FPR) and false negative rate (FNR). True positives were defined as PVs predicted as [likely]

deleterious and true negatives as BVs predicted as benign. False positives were defined as BVs predicted as [likely] deleterious, while false negatives were PVs predicted as benign. Additionally, the Matthews Correlation Coefficient (MCC) was calculated for all VPPTs using Eq. 1.

$$
\begin{aligned}
Sensitivity &= \tfrac{TP}{TP+FN} \\
Specificity &= \tfrac{TN}{TN+FP} \\
FPR &= \tfrac{FP}{TN+FP} \\
FNR &= \tfrac{FN}{TP+FN} \\
MCC &= \tfrac{TP \times TN - FP \times FN}{\sqrt{(TP+FP) \times (TP+FN) \times (TN+FP) \times (TN+FN)}}
\end{aligned}
\tag{1}
$$

R Base Functions[25] and RStudio[26] (version: 2023.6.1.524) were used for data analysis and formula calculations.

After exclusion for known PVs and BVs, VUS were further defined using a data-driven workflow for potential deleterious variant (PDV) identification (Fig. 2). To further define PDV oncogenic potential, we applied the power of Cancer Genome Interpreter (CGI) (version: March 2024)[27] with additional inclusion of predicted Loss of Function (pLoF), retrieved from Ensembl's Variant Effect Predictor (release 112)[28].

**Reporting summary**

Further information on research design is available in the Nature Portfolio Reporting Summary linked to this article.

## Results

### African-vs-European ancestral assessment of VPPTs using the benchmark datasets

While the 50 African PCa cases present with 1.6- and 1.2-fold greater number of SNVs and rare variants, respectively, we found the 50 European PCa cases to present with 2.5- and 1.4-fold more known ClinVar classified positive PVs or negative BVs, respectively (Table 1). Although ACMG-AMP guidelines (using InterVar) provided a better balance between the ancestries, we observed a 1.1-fold increase in positive PVs called for European men, with this variance widening to 2.8-fold for negative BVs. Notably, all 54 VPPTs were predicted with slightly higher mean sensitivity

## Table 2 | Summary of VPPTs tested in the study

| Tool | Cut-off | Year developed | Prediction model method |
|---|---|---|---|
| SIFT[53] | * | 2001 | MSA |
| bStatistics[54] | ≥500 | 2009 | MSA |
| LRT[55] | * | 2009 | MSA |
| MutPred[56] | ≥0.5 | 2009 | MSA + Protein parameters + Supervised ML (built on SIFT) |
| SiPhy[57] | ≥12.17 | 2009 | MSA |
| phastCons470way-mammalian[58] | ≥0.5 | 2010 | MSA |
| phastCons17way-primate[58] | ≥0.5 | 2010 | MSA |
| phastCons100way-vertebrate[58] | ≥0.5 | 2010 | MSA |
| phyloP470way-mammalian[58] | ≥1.6 | 2010 | MSA |
| phyloP17way-primate[58] | ≥1.6 | 2010 | MSA |
| phyloP100way-vertebrate[58] | ≥1.6 | 2010 | MSA |
| PolyPhen2-DIV[59] | * | 2010 | MSA + Protein parameters + Supervised ML |
| PolyPhen2-VAR[59] | * | 2010 | MSA + Protein parameters + Supervised ML |
| MutationAssessor[60] | * | 2011 | MSA |
| PROVEAN[61] | * | 2012 | MSA |
| FATHMM[62] | * | 2013 | MSA |
| VEST4[63] | ≥0.5 | 2013 | Supervised ML |
| CADD[64] | >20 | 2014 | Meta-predictor + Supervised ML + Unsupervised ML |
| MutationTaster[65] | * | 2014 | MSA |
| DANN[66] | ≥0.99 | 2015 | Supervised ML + DL (with the same training data set and features as CADD) |
| fathmm-MKL[67] | * | 2015 | Supervised ML |
| GenoCanyon[68] | ≥0.5 | 2015 | Unsupervised ML |
| GM12878_fitcons[69] | ≥0.6 | 2015 | Unsupervised ML |
| h1_fitcons[69] | ≥0.6 | 2015 | Unsupervised ML |
| HUVEC_fitcons[69] | ≥0.6 | 2015 | Unsupervised ML |
| integrated_fitcons[69] | ≥0.6 | 2015 | Unsupervised ML |
| MetaLR[70] | * | 2015 | Meta-predictor + Supervised ML |
| MetaSVM[70] | * | 2015 | Meta-predictor + Supervised ML |
| BayesDel_addAF[71] | * | 2016 | Meta-predictor + Supervised ML |
| BayesDel_noAF[71] | * | 2016 | Meta-predictor + Supervised ML |
| Eigen-PC[46] | ≥0 | 2016 | Unsupervised ML |
| Eigen-raw[46] | ≥0 | 2016 | Unsupervised ML |
| M-CAP[72] | * | 2016 | Supervised ML |
| REVEL[73] | ≥0.5 | 2016 | Meta-predictor + Supervised ML |
| SIFT-4G[74] | * | 2016 | MSA |
| DEOGEN2[75] | * | 2017 | Supervised ML |
| LINSIGHT[76] | ≥0.6 | 2017 | Unsupervised ML |
| MPC[77] | ≥2 | 2017 | MSA (also combined PolyPhen2) |

## Table 2 (continued) | Summary of VPPTs tested in the study

| Tool | Cut-off | Year developed | Prediction model method |
|---|---|---|---|
| ClinPred[78] | * | 2018 | Meta-predictor + Supervised ML + Unsupervised ML |
| fathmm-XF[79] | * | 2018 | Supervised ML |
| PrimateAI[80] | * | 2018 | Supervised ML + DL |
| GERP-NR[81] | ≥4 | 2020 | MSA |
| GERP-RS[81] | ≥4 | 2020 | MSA |
| LIST-S2[82] | * | 2020 | MSA |
| EVE[83] | ≥0.5 | 2021 | Unsupervised ML |
| MVP[84] | ≥0.75 | 2021 | Supervised ML + DL |
| VARITY-ER[85] | ≥0.5 | 2021 | Supervised ML |
| VARITY-ER-LOO[85] | ≥0.5 | 2021 | Supervised ML |
| VARITY-R[85] | ≥0.5 | 2021 | Supervised ML |
| VARITY-R-LOO[85] | ≥0.5 | 2021 | Supervised ML |
| gMVP[86] | ≥0.75 | 2022 | Supervised ML |
| MetaRNN[87] | * | 2022 | Meta-predictor + Supervised ML + DL |
| AlphaMissense[88] | * | 2023 | Unsupervised ML |
| ESM1b[89] | * | 2023 | Unsupervised ML + DL |

Tools with the directly predicted classification of variant pathogenicity were marked as *.
*DL* deep learning, *MSA* multiple sequence alignment, *ML* machine learning.

for European variants (0.68 *vs* 0.70, $p = 0.026$) derived from the merged ClinVar and InterVar benchmark datasets (Supplementary Table 1). Irrespective of ancestry (African *vs* European), VPPTs with the greatest sensitivity include M-CAP (0.98 *vs* 0.96), GERP-NR (0.92 *vs* 0.94) and CADD (0.91 *vs* 0.97), while GERP-NR had the highest FPR (0.81 *vs* 0.80). The greatest sensitivity differences between the ancestries included ClinPred preferencing African variants (0.67 *vs* 0.48), while VARITY-ER-LOO outperformed with European variants (0.63 *vs* 0.49). However, taken together, the 54 VPPTs predicted slightly better with African than European variants (mean MCC 0.19 *vs* 0.17, $p = 8.57E\text{-}06$). VPPTs with top overall performance included BayesDel-addAF, MetaRNN, ClinPred, LINSIGHT and BayesDel-noAF, with only LINSIGHT outperforming in the European over African cohort (0.52 *vs* 0.46). The raw number of positive and negative variants predicted is available in Supplementary Table 2.

Given the lack of pathogenic representation for African-derived variants within ClinVar and in turn bias of our benchmark dataset towards European-derived variants of known pathogenicity (both positive and negative), we sought to independently test VPPT performance by ancestry for ClinVar classified and InterVar ACMG-AMP guideline predicted variants. For known ClinVar variants, we found the predicting performance to be similar between our African and European cohorts (MCC 0.13 *vs* 0.14, $p = 3.19E\text{-}02$) (Supplementary Table 3), while sensitivity was marginally higher for African variants (0.77 *vs* 0.74, $p = 0.063$). Independent of ancestry, top-performing tools overall included MetaRNN (0.61 African *vs* 0.51 European), ClinPred (0.60 *vs* 0.49), BayesDel-addAF (0.54 *vs* 0.50), BayesDel-noAF (0.28 *vs* 0.28) and REVEL (0.25 *vs* 0.26). The raw number of positive and negative variants predicted is available in Supplementary Table 4. In contrast and as predicted, the sensitivity for African pathogenicity prediction declined when restricting our analyses to the InterVar dataset (Table 3, Supplementary Fig. 1), significantly increasing the ancestry gap (0.66 *vs* 0.71, $p = 9.86E\text{-}06$). Only 13 VPPTs (24.1%, 13/54) outperformed with higher sensitivity in our African versus European data, compared to 16 VPPTs (29.6%, 16/54) in the merged benchmark dataset. For Africans, the top five tools with the highest sensitivity in descending order included M-CAP, MetaSVM, MetaLR, GERP-NR and phastCons470way-mammalian, while for Europeans these included M-CAP,

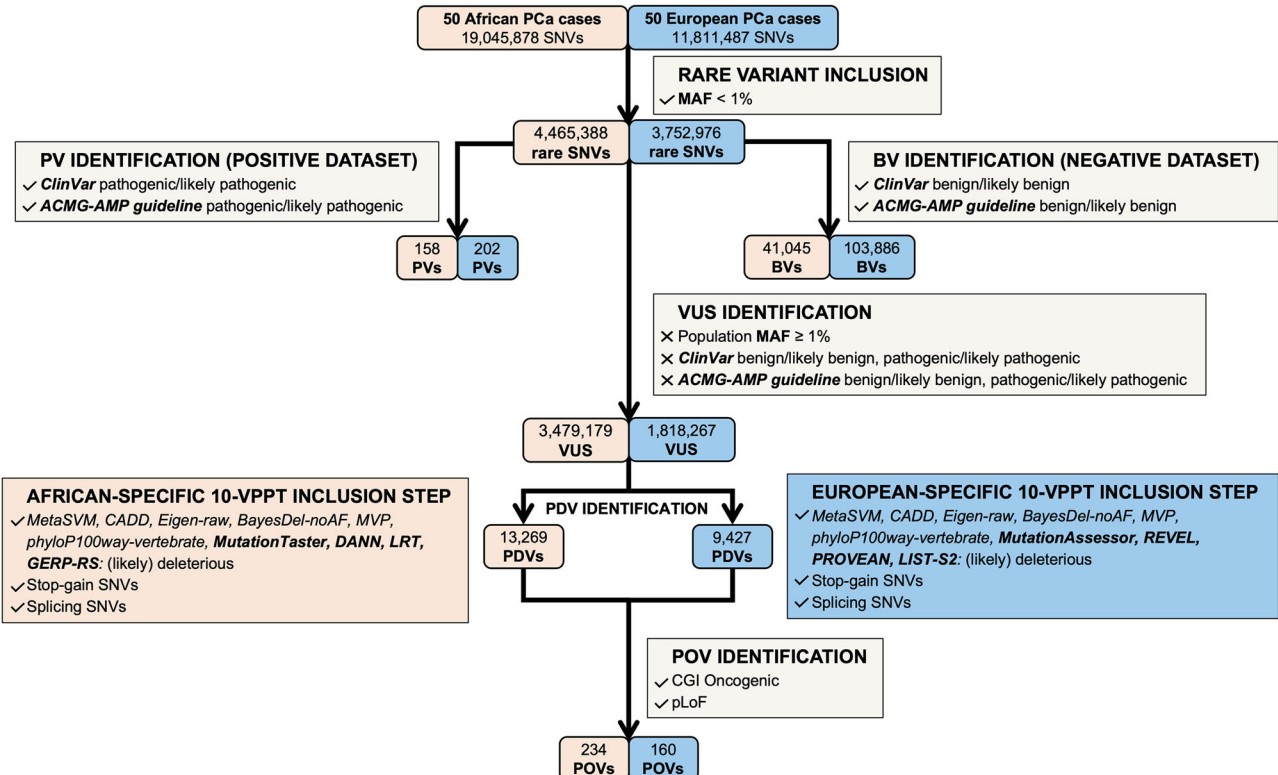

**Fig. 2 | Pathogenic prediction workflow for whole genome single-nucleotide variant (SNV) interrogation for 50 African (orange) and 50 European (blue) clinically matched advanced prostate cancer cases.** Rare SNVs were further defined as pathogenic variants (PVs) or benign variants (BVs) to establish the positive and negative benchmark variant datasets, respectively. Variants of unknown significance (VUS) were further interrogated for classification of potentially deleterious variants (PDVs) using ancestral-specific 10-VPPT criteria, with ancestral-unique VPPTs in bold, with further cancer-specific classification as potentially oncogenic variants (POVs), based on further interrogation for potential Loss of Function (pLoF) and Cancer Genome Interpreter (CGI) oncogenic status.

CADD, GERP-NR, Eigen-raw and MutationTaster. Notably, GERP-NR accounted for the highest FPR in both the African and European cohorts (0.78 vs 0.77). VPPTs that favoured European variant prediction for sensitivity and showed the greatest ancestral disparity included VARITY-ER-LOO (0.45 vs 0.65), MutationAssessor (0.69 vs 0.87) and PROVEAN (0.62 vs 0.80). As to the overall performance, a slightly higher MCC score was still observed for the African dataset (0.23 vs 0.20, p = 7.91E-05). Irrespective of ancestry, the top-performing tools included BayesDel-addAF (0.74 vs 0.73), LINSIGHT (0.50 vs 0.67), MetaRNN (0.67 vs 0.64), ClinPred (0.61 vs 0.62), BayesDel-noAF (0.52 vs 0.46) and MetaSVM (0.51 vs 0.38), while they were among VPPTs with the lowest FPR, except LINSIGHT (0.30 vs 0.20). The raw number of positive and negative variants predicted is available in Supplementary Table 5.

**African-vs-European ancestral assessment for top-performing VPPTs**

Based on the MCC scores of 54 VPPTs tested with African and European variant datasets, we propose and tested ancestrally independent pathogenic prediction workflows, with additional relevance for calling cancer-specific pathogenicity (Fig. 2). Applied to our data, after exclusion for PVs and BVs, including exclusion for rare variants with a reported MAFs > 1% across the globally representative gnomAD v4.1 database, a total of roughly 5.3 million VUS (3,479,179 African and 1,818,267 European) remained. Based on our InterVar ACMG-AMP guideline benchmarking dataset derived VPPT performance, with FPR > 0.3 exclusion, we select the top 10 performed VPPTs by ancestry, proposing ancestry-specific 10-VPPT PDV detection criteria. While MetaSVM, CADD, Eigen-raw, BayesDel-noAF, phyloP100way-vertebrate and MVP are shared between the ancestral workflows, MutationTaster, DANN, LRT and GERP-RS are African-specific and MutationAssessor, PROVEAN, LIST-S2 and REVEL are

European-specific. The European-specific VPPTs outperformed for sensitivity between 1.13- and 1.29-fold over African data. Irrespective of patient ancestry, we further refine for PDVs through the inclusion of stop-gain and splicing variants as defined by ACMG-AMP guidelines[11], classifying 13,269 African and 9,427 European-derived VUS as potentially deleterious. While overall this represents a greater percentage of European (0.52%) over African (0.38%) VUS, as a proportion of the total number of rare genome-wide SNVs by ancestry, PDVs represent a 1.18-fold increase for our African data.

To provide further verification for our African versus European 10-VPPT PDV prediction, we assessed our datasets using the alternative ancestral workflow. While the number of identified PDVs decreased for our African cohort by 2.5% (338 variants), in contrast, the PDV count increased by 1.3% (120 variants) for the European data. Assuming these differences are largely driven by the four ancestrally unique VPPTs, further targeted assessment for these VPPTs using the alternative ancestral workflow showed a major decline for our African data by 19.6% (2367 PDVs) and a 19.5% (1435 PDVs) increase for our Europeans. Additionally, we observed a 1.2-fold increase in the FPR between the African and the European 10-VPPTs for our European data (0.2 vs 0.17) and a further 1.5-fold increase between the African and the European unique 4-VPPTs (0.25 vs 0.17). Assuming that VPPTs largely favour European-derived variant prediction, we focussed on the most commonly used VPPTs[20], namely PolyPhen2, SIFT, CADD and MutationTaster, with PolyPhen2, SIFT and MutationTaster further highlighted in the ACMG-AMP guidelines[11]. Using the African ancestral InterVar ACMG-AMP guideline benchmark dataset, further PolyPhen2 and SIFT tool-specific selection was based on the highest sensitivity with FPR < 0.3, identifying SIFT-4G and PolyPhen2-HVAR. Compared to the number of PDVs predicted using these four most common VPPTs, our ancestral-specific workflows increased the number of African

**Table 3 | Performance of 54 VPPTs for ancestry-specific African (AFR) vs European (EUR) advanced prostate cancer benchmark datasets classified by InterVar (ACMG-AMP guidelines) and ordered in descending order of sensitivity (Sen) for our African ancestral patients**

| Tool | Sen AFR | Sen EUR | Spec AFR | Spec EUR | FPR AFR | FPR EUR | FNR AFR | FNR EUR | MCC AFR | MCC EUR |
|---|---|---|---|---|---|---|---|---|---|---|
| M-CAP | 0.98 | 0.97 | 0.65 | 0.65 | 0.35 | 0.35 | 0.02 | 0.03 | 0.30 | 0.29 |
| MetaSVM | 0.95 | 0.89 | 0.96 | 0.95 | 0.04 | 0.05 | 0.05 | 0.11 | 0.51 | 0.38 |
| MetaLR | 0.94 | 0.88 | 0.94 | 0.94 | 0.06 | 0.06 | 0.06 | 0.13 | 0.43 | 0.34 |
| GERP-NR | 0.91 | 0.93 | 0.22 | 0.23 | 0.78 | 0.77 | 0.09 | 0.07 | 0.06 | 0.06 |
| phastCons470way-mammalian | 0.91 | 0.89 | 0.64 | 0.60 | 0.36 | 0.40 | 0.09 | 0.11 | 0.20 | 0.14 |
| CADD | 0.90 | 0.96 | 0.79 | 0.75 | 0.21 | 0.25 | 0.10 | 0.04 | 0.30 | 0.24 |
| MutationTaster | 0.90 | 0.92 | 0.71 | 0.69 | 0.29 | 0.31 | 0.10 | 0.08 | 0.22 | 0.16 |
| GenoCanyon | 0.89 | 0.88 | 0.30 | 0.31 | 0.70 | 0.69 | 0.11 | 0.12 | 0.08 | 0.06 |
| phastCons100way-vertebrate | 0.87 | 0.89 | 0.63 | 0.60 | 0.37 | 0.40 | 0.13 | 0.11 | 0.19 | 0.15 |
| Eigen-raw | 0.86 | 0.92 | 0.83 | 0.80 | 0.17 | 0.20 | 0.14 | 0.08 | 0.33 | 0.26 |
| fathmm-MKL | 0.86 | 0.87 | 0.68 | 0.65 | 0.32 | 0.35 | 0.14 | 0.13 | 0.21 | 0.16 |
| phyloP470way-mammalian | 0.85 | 0.88 | 0.67 | 0.63 | 0.33 | 0.37 | 0.15 | 0.12 | 0.20 | 0.16 |
| Eigen-PC-raw | 0.83 | 0.87 | 0.82 | 0.79 | 0.18 | 0.21 | 0.17 | 0.13 | 0.30 | 0.23 |
| LINSIGHT | 0.80 | 0.90 | 0.70 | 0.80 | 0.30 | 0.20 | 0.20 | 0.10 | 0.50 | 0.67 |
| phyloP100way-vertebrate | 0.79 | 0.82 | 0.75 | 0.70 | 0.25 | 0.30 | 0.21 | 0.18 | 0.22 | 0.17 |
| bStatistic | 0.78 | 0.66 | 0.20 | 0.20 | 0.80 | 0.80 | 0.22 | 0.34 | −0.01 | −0.05 |
| phastCons17way-primate | 0.78 | 0.79 | 0.57 | 0.55 | 0.43 | 0.45 | 0.22 | 0.21 | 0.13 | 0.10 |
| DANN | 0.77 | 0.76 | 0.77 | 0.74 | 0.23 | 0.26 | 0.23 | 0.24 | 0.23 | 0.16 |
| BayesDel-noAF | 0.76 | 0.89 | 0.96 | 0.94 | 0.04 | 0.06 | 0.24 | 0.11 | 0.52 | 0.46 |
| LRT | 0.76 | 0.72 | 0.80 | 0.77 | 0.20 | 0.23 | 0.24 | 0.28 | 0.24 | 0.16 |
| Polyphen2-HDIV | 0.74 | 0.85 | 0.68 | 0.66 | 0.32 | 0.34 | 0.26 | 0.15 | 0.12 | 0.11 |
| SIFT | 0.74 | 0.82 | 0.68 | 0.67 | 0.32 | 0.33 | 0.26 | 0.18 | 0.12 | 0.11 |
| GERP-RS | 0.73 | 0.75 | 0.83 | 0.79 | 0.17 | 0.21 | 0.27 | 0.25 | 0.26 | 0.19 |
| MVP | 0.73 | 0.80 | 0.88 | 0.87 | 0.12 | 0.13 | 0.27 | 0.20 | 0.24 | 0.27 |
| SIFT-4G | 0.73 | 0.76 | 0.76 | 0.73 | 0.24 | 0.27 | 0.27 | 0.24 | 0.15G | 0.12 |
| LIST-S2 | 0.70 | 0.79 | 0.83 | 0.80 | 0.17 | 0.20 | 0.30 | 0.21 | 0.18 | 0.15 |
| MutationAssessor | 0.69 | 0.87 | 0.77 | 0.75 | 0.23 | 0.25 | 0.31 | 0.13 | 0.15 | 0.15 |
| BayesDel-addAF | 0.68 | 0.78 | 0.99 | 0.99 | 0.01 | 0.01 | 0.32 | 0.22 | 0.74 | 0.73 |
| ClinPred | 0.67 | 0.58 | 0.99 | 1.00 | 0.01 | 0.00 | 0.33 | 0.42 | 0.61 | 0.62 |
| SiPhy | 0.67 | 0.76 | 0.82 | 0.80 | 0.18 | 0.20 | 0.33 | 0.24 | 0.23 | 0.20 |
| Polyphen2-HVAR | 0.65 | 0.77 | 0.79 | 0.77 | 0.21 | 0.23 | 0.35 | 0.23 | 0.15 | 0.14 |
| REVEL | 0.65 | 0.78 | 0.97 | 0.95 | 0.03 | 0.05 | 0.35 | 0.22 | 0.39 | 0.33 |
| ESM1b | 0.64 | 0.73 | 0.86 | 0.84 | 0.14 | 0.16 | 0.36 | 0.27 | 0.19 | 0.16 |
| FATHMM | 0.64 | 0.76 | 0.83 | 0.85 | 0.17 | 0.15 | 0.36 | 0.24 | 0.17 | 0.17 |
| PROVEAN | 0.62 | 0.80 | 0.83 | 0.81 | 0.17 | 0.19 | 0.38 | 0.20 | 0.16 | 0.16 |
| MutPred | 0.59 | 0.73 | 0.79 | 0.80 | 0.21 | 0.20 | 0.41 | 0.27 | 0.20 | 0.20 |
| VEST4 | 0.59 | 0.73 | 0.89 | 0.90 | 0.11 | 0.10 | 0.41 | 0.27 | 0.24 | 0.26 |
| fathmm-XF | 0.58 | 0.54 | 0.86 | 0.84 | 0.14 | 0.16 | 0.42 | 0.46 | 0.20 | 0.13 |
| MetaRNN | 0.56 | 0.49 | 1.00 | 1.00 | 0.00 | 0.00 | 0.44 | 0.51 | 0.67 | 0.64 |
| VARITY-R | 0.55 | 0.68 | 0.94 | 0.93 | 0.06 | 0.07 | 0.45 | 0.32 | 0.28 | 0.23 |
| VARITY-R-LOO | 0.55 | 0.68 | 0.93 | 0.92 | 0.07 | 0.08 | 0.45 | 0.32 | 0.26 | 0.23 |
| DEOGEN2 | 0.54 | 0.66 | 0.95 | 0.94 | 0.05 | 0.06 | 0.46 | 0.34 | 0.28 | 0.25 |
| hESC-fitCons | 0.53 | 0.55 | 0.35 | 0.38 | 0.65 | 0.62 | 0.47 | 0.45 | −0.04 | −0.02 |
| HUVEC-fitCons | 0.53 | 0.45 | 0.40 | 0.41 | 0.60 | 0.59 | 0.47 | 0.55 | −0.03 | −0.04 |
| integrated-fitCons | 0.53 | 0.50 | 0.40 | 0.42 | 0.60 | 0.58 | 0.47 | 0.50 | −0.03 | −0.02 |
| EVE | 0.48 | 0.61 | 0.84 | 0.82 | 0.16 | 0.18 | 0.52 | 0.39 | 0.18 | 0.20 |
| VARITY-ER | 0.45 | 0.61 | 0.96 | 0.94 | 0.04 | 0.06 | 0.55 | 0.39 | 0.26 | 0.23 |
| VARITY-ER-LOO | 0.45 | 0.65 | 0.96 | 0.94 | 0.04 | 0.06 | 0.55 | 0.35 | 0.25 | 0.25 |
| gMVP | 0.41 | 0.56 | 0.96 | 0.94 | 0.04 | 0.06 | 0.59 | 0.44 | 0.22 | 0.21 |

**Table 3 (continued) | Performance of 54 VPPTs for ancestry-specific African (AFR) vs European (EUR) advanced prostate cancer benchmark datasets classified by InterVar (ACMG-AMP guidelines) and ordered in descending order of sensitivity (Sen) for our African ancestral patients**

| Tool | Sen AFR | Sen EUR | Spec AFR | Spec EUR | FPR AFR | FPR EUR | FNR AFR | FNR EUR | MCC AFR | MCC EUR |
|---|---|---|---|---|---|---|---|---|---|---|
| GM12878-fitCons | 0.38 | 0.44 | 0.54 | 0.56 | 0.46 | 0.44 | 0.62 | 0.56 | −0.03 | 0.00 |
| AlphaMissense | 0.37 | 0.53 | 0.97 | 0.95 | 0.03 | 0.05 | 0.63 | 0.47 | 0.23 | 0.21 |
| PrimateAI | 0.20 | 0.20 | 0.98 | 0.98 | 0.02 | 0.02 | 0.80 | 0.80 | 0.15 | 0.12 |
| MPC | 0.03 | 0.03 | 1.00 | 1.00 | 0.00 | 0.00 | 0.97 | 0.97 | 0.05 | 0.04 |
| phyloP17way-primate | 0.00 | 0.00 | 1.00 | 1.00 | 0.00 | 0.00 | 1.00 | 1.00 | 0.00 | 0.00 |
| Mean | 0.66 | 0.71 | 0.77 | 0.76 | 0.23 | 0.24 | 0.34 | 0.29 | 0.23 | 0.20 |
| *p* value | *9.86E-06* | | *2.54E-03* | | *2.54E-03* | | *9.86E-06* | | *7.91E-05* | |

*FNR* false negative rate, *FPR* false positive rate, *MCC* Matthew's correlation coefficient, *Sen* sensitivity, *Spec* specificity.

predicted PDVs by 1.2% (159 PDVs) and only minimally decreased the European predicted PDVs by 0.15% (14 PDVs).

### Assessment of ancestry-specific 10-VPPT workflows to predict oncogenicity

To further evaluate our 10-VPPT ancestry-specific criteria to improve cancer-specific studies, we propose additional filtering steps to refine potentially PVs as potentially oncogenic variants (POVs). Here we utilise the power of predicting pLoF variants, as well as identifying oncogenic genetic alterations using CGI[27]. Classically, LoF variants are regarded as gene-disrupting alterations and result in a premature stop codon for protein transcription[29], while oncogenic variants often occur upstream and cause tumour suppressor inactivation[30]. Identifying 23 African and 9 European pLoFs, none overlapped with CGI predictions for a merged total of 234 and 160 POVs, respectively. Notably, our POVs increased the per capita ClinVar/InterVar PVs (Number of PVs/Number of patients) from 3.16 for Africans and 4.04 for Europeans to 7.84 and 7.24, respectively. As a total number of genome-wide SNVs, the gap between POV prediction equated to a 1.1-fold increase for African over European data.

### Discussion

Although VPPTs have been widely utilised and evaluated for specific diseases[31] and variant types[32,33], most VPPT comparison studies have excluded African ancestral data[34–36]. Focusing on an African-*vs*-European ancestral whole genome comparative PCa cohort including men of Southern African versus European ancestries, our study highlights the need to ensure that VPPTs are appropriate for African inclusion, as we demonstrated the inadequacies of current pathogenic databases and prediction guidelines. Importantly, while our patients were clinicopathologically matched, cases were further excluded for genetic admixture, while using a single data generation and variant calling pipeline, we excluded for inevitable between-study variability[18]. At the epicentre of human genome diversity[9,37,38], it is not surprising that our southern African patients present with 1.6- and 1.2-fold more genome-wide SNVs and rare SNVs. Conversely, European patients have a greater percentage of rare variants (31.8% *vs* 23.5%), which may be explained through recently diverged populations experiencing a major genetic bottleneck during the out-of-Africa migration event[39,40]. Irrespective, the proportion of PVs and BVs is double to four times that for European over African SNVs, respectively, emphasising the associated discrepancies with regard to ClinVar content and InterVar ACMG-AMP guidelines for African inferences. Using our established positive and negative benchmark datasets, we evaluated VPPT performance, suggesting a 10-VPPT African-specific workflow to maximise PDV identification. Using these criteria, we increased the proportion of PDVs identified in our African data and in turn narrowed the margin observed between the ancestries. As our study focuses on PCa, through further loss of function and oncogenic predictions, we provide further clarification for potential pathogenicity, again narrowing the ancestral prediction gap.

A thorough review of the literature showed a single study to include African ancestral data for VPPT assessment[32]. Focusing on African American data and as such representing predominantly West African genomic ancestry, we and others have shown genetic substructure fraction divergence between African Americans and Southern Africans[41,42]. However, similarities between the studies are noted. These include a limited gap of specificity between African and European ancestral data, while the pattern of specificity is higher for LRT, CADD and MutationTaster and lower for FATHMM for both African American and Southern African data. In contrast, Southern African men did not show higher specificity for VEST4 and PolyPhen2, nor lower specificity for PROVEAN, MutationAssessor and SIFT. Moreover, we showed current VPPTs performed worse when predicting for African PVs, while the disparity of sensitivity was greater for the InterVar ACMG-AMP guideline benchmark datasets, compared to the merged benchmark datasets, suggesting current VPPTs were trained biased towards the ClinVar database. Given the difference in the inter-ethnic gap of sensitivity and specificity of current VPPTs, we infer whether the prediction of rare deleterious variants is more computationally challenging than rare non-functional/neutral variants.

Based on our findings, we provide further discussion with regards to ancestry-specific VPPT selection. Among the six ancestrally shared top-performing VPPTs, three are meta-predictors, including MetaSVM, CADD and BayesDel-noAF with the unsupervised ML VPPT Eigen-raw, the supervised MVP with DL and the MSA VPPT phyloP100way-vertebrate. Meta-predictors such as BayesDel, CADD, MetaSVM, ClinPred and REVEL generally have better performance[43], while MetaSVM has some of the lowest FPR[44]. Eigen and phyloP have similarly shown decent performance for sensitivity and FPR[34]. Three MSA VPPTs each were selected as African- and European-specific, with the supervised DANN complimenting the African workflow and the Meta-predictor REVEL the European workflow. Irrespective of ancestry, no VPPT relying solely on supervised ML made our selection criteria due to either low sensitivity or high FPRs. The concern was raised for supervised ML that the same set of variants is recursively used for both training and performance assessment, resulting in poor prediction on unknown data[19,45]. Although several VPPTs performed with very high sensitivity, such as M-CAP, GERP-NR and GenoCanyon, they were excluded for consideration due to high FPRs. Notably, two independent European-based comparison studies found M-CAP, followed by GenoCanyon, to have one of the highest FPRs[35,44]. Further consideration must also be given to VPPTs designed for specific variant types. For instance, as LINSIGHT has been built to predict non-coding variants only[46], it is not surprising that it generated the least number of predicted positive and negative variants.

Since ACMG recommended for the return of genomic incidental findings from a minimum set of 56 actionable genes in clinical sequencing[47], large-scale studies[48,49] have sought to define and standardise the criteria for variant pathogenicity classification, including variant allele frequency, segregation, the number of patients affected, de novo events in a trio and disruptive variants, while VPPTs were tested for its utility although they

were not involved in their criteria. Their results showed a lower (by nearly 50%) number of African PVs in actionable genes than European ancestry, where the authors concluded that this could be attributed to the under-representation of the African participants in the publicly available databases and literature. Furthermore, a perspective article from Southern African researchers raised the utility of VPPTs in African genomes and called for an approach to infer African variant pathogenicity[15]. While our focus on Southern Africans provides advantages with respect to minimising non-African and non-Southern African genetic admixture, in turn, we acknowledge the limitation of our observations to be translated across the African diaspora, calling for further regionally relevant and ethnolinguistically inclusive studies. Furthermore, our and other studies are limited by a lack of disease-negative population-matched control resources. This has led to the generation of public data such as the Australian Medical Genome Reference Bank, which provides the community with whole genome variant and associated phenotype data for over 3000 healthy aged (>75 years) participants with no known cancer, cardiovascular disease or dementia[50]. However, to the best of our knowledge, no such resource exists for Sub-Saharan Africa. In contrast, while resources like gnomAD capture WGS data for 37,545 African genomes (gnomAD v4)[51], phenotypic data are not prioritised and as previously demonstrated, African representation is regionally confined and lacking for Southern African inclusion[42]. Here we advocate for the establishment of younger-aged disease-free population-matched resources. Our rationale, as human aging increases the frequency of somatic mutations in hematopoietic stem cells, a phenomenon more prevalent in DNA repair genes, while the risk is modest, their rarity has the potential for further misclassification for germline screening[52]. Either way, accurate filtering for non-pathogenic African-relevant PDVs requires a substantial global commitment.

## Conclusion

Here we curate, to the best of our knowledge, the first ancestry-specific set of VPPTs informed by our African-vs-European ancestral genome-wide variant evaluation. Establishing African-specific criteria, we begin to close the ancestry gap for PV prediction using tailored in silico methods. While just the beginning, our study is a call for action—our results highlighting the need for further representation across the broad African identifier, further disease type-specific evaluations, and establishing African-relevant and regionally specific databases from clinically curated to disease-free resources.

## Data availability

The sequence data were made accessible through the European Genome-Phenome Archive (EGA) accession EGAS00001006425 and derived from the Southern African Prostate Cancer Study (SAPCS) Dataset (EGAD00001009067; https://www.ega-archive.org/datasets/EGAD000010 09067) and European Australian Garvan/St Vincent's Prostate Cancer Dataset (EGAD00001009066; https://www.ega-archive.org/datasets/EGA D00001009066).

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

## Acknowledgements

We are grateful to our colleagues from the Ancestry and Health Genomics Laboratory at the University of Sydney, specifically Jue Jiang, for high-throughput data processing and Dr Weerachai Jaratlerdsiri for stimulating discussions, as well as the patients, clinical staff, the National Health and Medical Research Council (NHMRC) of Australia (APP1165762, APP2001098 and APP2010551 to V.M.H.) for funding the genomic sequencing, and the many authors who contributed to the published works in Jaratlerdsiri et al.[18] We would like to specifically acknowledge our co-principal leads of the DoD-funded HEROIC PCaPH Africa1K Consortium, including SAPCS co-Director Professor M.S. Riana Bornman from the University of Pretoria (South Africa), as well as Professors Peter Mungai Ngugi from the University of Nairobi (Kenya) and Gail S. Prins from the University of Illinois at Chicago (USA). Financial support for analytics was provided by the USA Congressionally Directed Medical Research Programs (CDMRP) DoD Prostate Cancer Research Program (PCRP) via a HEROIC Consortium Award (PC210168 and PC230673, HEROIC PCaPH Africa1K to V.M.H.), a USA National Institute of Health (NIH) National Cancer Institute (NCI) Award (1R01CA285772-01 to V.M.H.) and a USA Prostate Cancer Foundation (PCF) Challenge Award (2023CHAL4150 to V.M.H.). K.Z. is supported by a University of Sydney International Scholarship and V.M.H. by the Petre Foundation via the University of Sydney Foundation, Australia. This work will form part of a Ph.D. thesis for K.Z.

## Author contributions

Conception and methodology: all authors; formal analysis: K.Z.; writing and figures: K.Z.; supervision, review and editing: K.G., P.X.Y.S. and V.M.H.; funding acquisition: V.M.H.

## Competing interests

V.M.H. is a Member of the Active Surveillance Movember Committee and received an honorarium from the Korean Urological Oncology Society for the 2024 Annual Conference as a guest speaker. Authors K.Z., K.G. and P.X.Y.S. declare no competing interests.
