## [Transparent Peer Review file · Communications Medicine]

Evaluating variant pathogenicity prediction tools to establish African inclusive guidelines for germline genetic testing

Corresponding Author: Professor Vanessa Hayes

Version 0:

Reviewer comments:

Reviewer #1

(Remarks to the Author)

The major goal of this paper is to establish a pathogenic variant prediction workflow for advanced prostate cancer patients with Southern African ancestry using available prediction tools. It suggests that top-performing prediction tools specific to this ancestry are effective for capturing ancestry-specific variants. This work is novel and will be of interest to clinician researchers and the precision medicine team for effective clinical use. It is thorough and convincing, and will be helpful to the broader community of researchers, clinicians, and health professionals. This type of paper is needed in the field, as data on African ancestry is very limited. Without this type of work, the gap in genetic health disparity cannot be reduced.

I have no concerns about the statistical validity of the work. As mentioned by the authors, the lack of effective controls without prostate cancer of African ancestry creates some limitations for the study. In the future, increasing the sample size of advanced prostate cancer cases with African ancestry and matched controls will be a significant next step.

I have a few minor suggestions on the effective communication of the paper. I have added an underline if that specific word needs to be inserted for better presentation:

1. Line 29. the use of the word 'hypothesis' not relevant when the hypothesis was not mentioned anywhere.
2. Line 34. "Using these guidelines..." should be better with the replacement of "Using these pathogenic prediction workflows.."
3. Line 49. "Using our guidelines.." with "Using our pathogenic prediction workflows.."
4. Line 117. "...non-admixed Southern African.."
5. Line 118. "...European Australian..."
6. Line 159. Table 1: Southern African European Australian; Mean age(range) years.
7. Line 242: Acc, accuracy should be removed; Sen and Spec should be added on the abbreviations list
8. Line 284. Replace 'Study' with 'Pathogenic prediction'
9. Line 285. "...clinically matched advanced prostate..."
10. Line 317. 'Cases' to be replaced by 'patients'

Reviewer #2

(Remarks to the Author)

The authors analysed whole genomes from 50 African and 50 European prostate cancer patients to evaluate the performance of 54 variant pathogenicity prediction tools (VPPTs).

The paper is very well written, analyses well done and well explained. The datasets are available upon standard data access approval, all the VPPTs tested are available, the raw data is provided, and the results are thus reproducible. Overall, the results highlight the need for ancestry specific pipelines for pathogenic variant prediction.

There has long been an acknowledgment within the genomic research community of the poor representation of non-European populations in major population genomic cohorts. This imbalance has led to significant gaps in the understanding of genetic diversity and its implications for health and disease in underrepresented populations. To the best of my knowledge, this study represents the first extensive effort to systematically assess the ancestry bias in most available VPPTs.

While the overall sensitivity of the tools was lower for the African data, the authors were able to identify which tools were suitable irrespective of ancestry and which ones performed better for the African/European data. Importantly, the results and proposed workflow will provide valuable guidelines for the research community in predicting pathogenic variants in South African patients. This work marks a significant step toward making germline testing more inclusive of underrepresented populations.

I have a few points for the authors' consideration:

1. While I appreciate that one of the datasets used is called the "Southern African Prostate Cancer Study" and the authors do show a clear awareness of the issues with labels, I would suggest being consistent throughout the paper and selecting one term, and not using "Southern African" (e.g. lines 24 and 113) and "South African" (e.g. line 133 and 138) interchangeably. The "Southern African" dataset includes 123 prostate cancer patients from South Africa and 7 from Brazil, and none from other Southern African countries (e.g. Namibia and Botswana). I do, however, appreciate the need to use the distinctions "African" versus "European" when discussing on a broader scale as the authors clearly acknowledge the issues with the "African" label given the high linguistic and genetic diversity within the continent.
2. I am slightly confused about the numbers of individuals given in lines 117-118 (Southern African (n = 113) and European (n = 54)), as I don't see how they match the data source recorded for the EGA repository (South Africa (n=123), Brazil (n=7) and Australia (n=53)). Was the data used in this paper a combination of EGA and additional data? If this is the case, please clarify in your description.
3. Is it perhaps a bit of a stretch to describe the dataset as "multi-ethnic" as it essentially consists of two populations (Australian European and South African)? Going back to comment #2 about the "Southern African" label, I wonder whether this also implies that the data represents more diversity than they have?
4. Given that the cohort was selected to match patients based on pathology and ancestry, would it be possible to display the results of the population substructure methods used to remove the admixed individuals (e.g. plot from fastSTRUCTURE in the supplement)?
5. Consider, making the difference in the mean PSA for African versus European men clearer in the sentence (line 136): Notably, PSA is excluded as an inclusion criteria as a consequence of the significant elevation in overall PSA levels at presentation in our African men (mean 448.3 versus 12.8 ng/mL [in the European men], $p = 0.001$),...
6. I wonder whether the box containing the PV/BV identification should state "InterVar" rather than "ACMG-AMP guideline" as InterVar is more frequently used elsewhere to describe the dataset (Table 1 and main text). And perhaps also refer to them as the "merged" set to match other references in the paper (Supplementary Tables and main text)?
7. Remove "for" from sentence in line 309.
...most VPPT comparison studies have excluded [for] African ancestral data69-71.
8. Supplementary document includes a comment on the side from one of the authors (by Supplementary Figure 1).

Reviewer #3

(Remarks to the Author)

The authors are conducting critical analyses to evaluate and improve the performance of Variant Pathogenicity Prediction Tools (VPPTs) across ancestries, a significant step toward addressing biases and enhancing prediction accuracy in diverse populations. This work is particularly important for identifying VPPTs that perform well in African-ancestry populations, who are often underrepresented in genomic studies. VPPTs may exhibit varying levels of accuracy for different classes of genetic variants. For instance, these tools often perform well with nonsynonymous variants that directly affect protein function but may struggle with splicing or regulatory variants due to the complexity of their effects on gene expression. A focused analysis of tool performance across variant classes could identify weaknesses and guide the development of more comprehensive VPPTs. Additionally, aggressive prostate cancer (CaP) may be driven by mechanisms tied to specific variant classes. Nonsynonymous variants might indicate disruptions in key cancer-related pathways, while splicing variants could contribute to dysregulated gene expression patterns unique to aggressive disease phenotypes.

Investigating rare variant burden—a measure of the cumulative effect of rare variants within a genome, gene, or pathway—could reveal whether individuals with aggressive CaP carry a higher load of pathogenic variants compared to controls. Rare variant analyses could also identify genes or pathways enriched for rare variants, particularly in African versus European populations. Furthermore, studying the contribution of rare variants to missing heritability could clarify their role in aggressive CaP by quantifying their impact on disease heritability and explaining ancestry-specific differences. This comprehensive approach would advance the understanding of aggressive CaP genetics, highlight the need for ancestry-specific methods, and potentially inform therapeutic strategies and personalized medicine efforts.

To further strengthen this study, the authors could explore whether common variants exhibit any differences in VPPT performance across ancestries. Common variants tend to have more robust functional annotations and are better characterized in existing genomic datasets. Investigating their predictive performance by ancestry could reveal whether the observed improvements in VPPTs are consistent across variant classes or are specific to rare or ancestry-enriched variants. This comparison could offer additional insights into the generalizability and limitations of current VPPTs and help identify opportunities for refining tools to address ancestry-specific genetic variation.

Reviewer #4

(Remarks to the Author)

The authors test the hypothesis that there exists ancestry-specific ariability in predictive power across algorithms designed for variant pathogenicity. In particular, the focus is on prostate cancer, where African individuals are less likely to have pathogenic variants but more likely to have disease. While the article is at times difficult to follow, the methodology is sound and the aim is valuable. A few suggestions:

1. In table 1 the authors should conduct a statistical test comparing the two groups across parameters mentioned. Are they significantly different in any regard, or within the normal range of variation.
2. In table 3 it is not clear the ordering of algorithms, they should be organized based on type of algorithm as described in Figure 1. Also it is not clear what are the major conclusions that can be drawn from this table.
3. While the authors do not present any code, I wonder if they implemented and can share a software pipeline that runs and compares the 54 algorithms, as this would be of great benefit to the community, in particular if it were in a stand-alone package or image that could be set up in a cloud computing environment.
4. In particular I am interested in understanding the author conclusions regarding AI-based algorithms, are they in general better or worse suited for African genome interpretation? What are the conclusions and recommendations of the authors?
5. While it appears to be out of scope, there would be tremendous value to run the same analysis on other cancer types. Is this phenomenon unique to prostate cancer? What about multifactorial diseases and pharmacogenomics variants?

Version 1:

Reviewer comments:

Reviewer #1

(Remarks to the Author)

The findings of this study will be important for effective clinical use. I have no further comments or concerns. All comments were addressed, and it is a well written paper.

Reviewer #2

(Remarks to the Author)

The authors have satisfactorily responded to all of my comments and no further changes are required.

Reviewer #3

(Remarks to the Author)

This paper evaluates variant pathogenicity prediction tools (VPPTs) in the context of African genomic diversity, addressing the bias towards European populations prevalent in current genetic testing and variant interpretation. Key findings include the lower sensitivity of current VPPTs for African-derived variants, highlighting the need for ancestry-specific workflows. The study identifies VPPTs that perform better in either African or European datasets, demonstrating consistent performance across ancestries. In contrast, MutationTaster, DANN, LRT, and GERP-RS perform better in African datasets, while MutationAssessor, PROVEAN, LIST-S2, and REVEL show higher performance in European datasets. By utilizing ancestry-specific 10-VPPT workflows, the study narrowed the gap in predicting potentially deleterious variants between African and European data. Furthermore, the paper presents a method for identifying potentially oncogenic variants by combining VPPT predictions with predicted Loss of function (pLoF) and Cancer Genome Interpreter (CGI) analysis. These findings are novel and will be of significant interest to the genomics community, particularly those focused on health equity and diverse populations. The study directly addresses the underrepresentation of African populations in genomic research, a critical issue hindering the equitable application of genomic medicine. The use of a matched cohort of Southern African and European men with advanced prostate cancer strengthens the study's validity. While acknowledging the need for larger and more diverse datasets, the study provides valuable insights into VPPT performance across ancestries, emphasizing the importance of ancestral diversity in genomic studies and advocating for the development of ancestry-specific guidelines for variant interpretation. This research will contribute to achieving health equity in genomic medicine by providing practical guidance for selecting VPPTs to maximize the prediction of rare pathogenic variants.

Reviewer #4

(Remarks to the Author)

The authors have produced a significantly improved manuscript that is easy to follow, presents the problem and discovers a solution. Some minor suggestions:

1. Please number the pages in the manuscript.
2. Fig 1. The earliest VPPTs predate the human genome project and were based on substitution matrices such as BLOSUM. Early manuscripts by Henikoff et al show applications in the 90's to individual gene sequences, these early methods evolved into SIFT and others. In parallel, methods based on protein structure prediction were developed, these also predate the Human genome project completion. I'd recommend updating this figure and the text describing it to reflect the early days of the field, where genes were sequenced and studied individually (without a whole genome sequence).

3. Table 3. I'm surprised by the poor reported performance of the "latest and greatest" algorithm ESM1b! Given that it is a recent and unique method, I would recommend some explanation in the manuscript results and discussion regarding this finding and what factors can be attributable for this performance of ESM1b. Based on what I know of ESM1b, the algorithm performs better on genes and gene regions where there is little MSA data available. Possibly the authors could look at variants and genes where in the ESM1b manuscript performance was reported to be higher, and confirm if this is the case for your African dataset.

Another way to sort this table and rate "performance" would be the difference in sensitivity or specificity between AF and EU populations. The main objective of this study is to identify and quantify those differences.

4. The abbreviations "AF" and "EU" are non-standard, I would suggest changing them to "AFR" and "EUR" in line with other publications such as 1000 Genomes Project, GNOMAD, etc.

5. Page 12, paragraph 1, line 266. I find the higher rate of VUS in European to be unexpected, can the authors elaborate on that contradictory finding? Likewise, the performance is higher for Afr variants (page 10 paragraph 1). Surprising finding, please elaborate in the discussion.

6. Page 13, line 280. How did the authors arrive at a determination of "most commonly used" algorithms? I would suggest making that criteria clear, such as # of citations and add this to Table 2.

7. Table 2. For clarity I would suggest adding columns to this table specifying which algorithms are ancestry-specific and other categories used in the results.

8.

Version 2:

Reviewer comments:

Reviewer #4

(Remarks to the Author)

The authors have addressed all of my questions and I wish them the best, hoping that this important research continues.

Reviewers' Comments and Author Response

Evaluating variant pathogenicity prediction tools to establish African inclusive guidelines for germline genetic testing.

Response to reviewer's: Blue

Referee #1: human evolutionary genetics and genomics, array comparative genomic hybridization using ddPCR

Reviewer #1 (Remarks to the Author):

The major goal of this paper is to establish a pathogenic variant prediction workflow for advanced prostate cancer patients with Southern African ancestry using available prediction tools. It suggests that top-performing prediction tools specific to this ancestry are effective for capturing ancestry-specific variants. This work is novel and will be of interest to clinician researchers and the precision medicine team for effective clinical use. It is thorough and convincing, and will be helpful to the broader community of researchers, clinicians, and health professionals. This type of paper is needed in the field, as data on African ancestry is very limited. Without this type of work, the gap in genetic health disparity cannot be reduced. I have no concerns about the statistical validity of the work. As mentioned by the authors, the lack of effective controls without prostate cancer of African ancestry creates some limitations for the study. In the future, increasing the sample size of advanced prostate cancer cases with African ancestry and matched controls will be a significant next step.

Response: We thank the reviewer for their comments and for recognising and acknowledging the tremendous gap and need for African-relevant genetic testing not only for all diseases, but of particular interest to our team's area of research – prostate cancer.

I have a few minor suggestions on the effective communication of the paper. I have added an underline if that specific word needs to be inserted for better presentation:

1. Line 29. the use of the word 'hypothesis' not relevant when the hypothesis was not mentioned anywhere.

Response: The words 'supports our hypothesis' have been removed.

2. Line 34. "Using these guidelines..." should be better with the replacement of "Using these pathogenic prediction workflows.."

Response: We replaced the word 'guidelines' with 'pathogenic prediction workflows' as suggested.

3. Line 49. "Using our guidelines.." with "Using our pathogenic prediction workflows.."

Response: We replaced the word 'guidelines' with 'pathogenic prediction workflows' as suggested.

4. Line 117. "...non-admixed Southern African.."

Response: The word 'southern' was added to reflect southern African heritage of the African patients. 'Southern' has been included at additional points across the manuscript.

5. Line 118. "...European Australian..."

Response: The word 'largely Australian' was added to reflect patient country of origin of the European patients who were most from Australia and included a single European ancestral South African (Table 1).

6. Line 159. Table 1: Southern African European Australian; Mean age(range) years.

Response: ethnicity was replaced by 'genetic ancestry' in the heading of Table 1 to better reflect that our reference is to the genetic ancestry rather than the country or region of the patients. This has also been further clarified in the main text under methods, from line 149 and reads "allowing for selection of patients representing exclusively 'African' or 'European' genetic ancestries." As such, we have not added 'Southern' or 'Australian' to the genetic ancestry headings in Table 1. We have added the text 'years' to accurately reflect age and 'ng/mL' to reflect PSA measurement.

7. Line 242: Acc, accuracy should be removed; Sen and Spec should be added on the abbreviations list

Response: Acc, accuracy was removed, Sen and Spec were present.

8. Line 284. Replace 'Study' with 'Pathogenic prediction'

Response: The word 'study' was replaced with 'pathogenic prediction' as suggested.

9. Line 285. "...clinically matched advanced prostate..."

Response: The word 'advanced' was added for further clarity as suggested.

10. Line 317. 'Cases' to be replaced by 'patients'

Response: 'Cases' was replaced with the word 'patients' as suggested.

Referee #2: Prediction of disease risks across multiple populations using evolutionary genetics/ bioinformatics

Reviewer #2 (Remarks to the Author):

The authors analysed whole genomes from 50 African and 50 European prostate cancer patients to evaluate the performance of 54 variant pathogenicity prediction tools (VPPTs). The paper is very well written, analyses well done and well explained. The datasets are available upon standard data access approval, all the VPPTs tested are available, the raw data is provided, and the results are thus reproducible. Overall, the results highlight the need for ancestry specific pipelines for pathogenic variant prediction.

Response: We appreciate the reviewers complements with regards to our study.

There has long been an acknowledgment within the genomic research community of the poor representation of non-European populations in major population genomic cohorts. This imbalance has led to significant gaps in the understanding of genetic diversity and its implications for health and disease in underrepresented populations. To the best of my knowledge, this study represents the first extensive effort to systematically assess the ancestry bias in most available VPPTs.

Response: We appreciate further recognition for the substantial knowledge gap and in turn the importance and significance of this work for the community.

While the overall sensitivity of the tools was lower for the African data, the authors were able to identify which tools were suitable irrespective of ancestry and which ones performed better for the African/European data. Importantly, the results and proposed workflow will provide valuable guidelines for the research community in predicting pathogenic variants in South African patients. This work marks a significant step toward making germline testing more inclusive of underrepresented populations.

Response: We appreciate the recognition of the importance of making germline testing a reality for all populations world-wide.

I have a few points for the authors' consideration:

1. While I appreciate that one of the datasets used is called the "Southern African Prostate Cancer Study" and the authors do show a clear awareness of the issues with labels, I would suggest being consistent throughout the paper and selecting one term, and not using "Southern African" (e.g. lines 24 and 113) and "South African" (e.g. line 133 and 138) interchangeably. The "Southern African" dataset includes 123 prostate cancer patients from South Africa and 7 from Brazil, and none from other Southern African countries (e.g. Namibia and Botswana). I do, however, appreciate the need to use the distinctions "African" versus "European" when discussing on a broader scale as the authors clearly acknowledge the issues with the "African" label given the high linguistic and genetic diversity within the continent.

Response: For clarification the sentence from line 136 was modified to read, "Representing the Southern African population identifier, our study included 50 Black South Africans (100% ISUP 4-5), while presenting the European identifier, our study included 49 Australian and 1 White South African (98% ISUP 4-5, with a single European man presenting with ISUP 3). This allows us to keep our reference to the Southern African Prostate Cancer Study (SAPCS), which is a known study that covers men from all countries represented within the region. To further clarify our genetic identifiers as African and European going forward in this paper, we have added some clarifying words in blue to the sentence that ends that paragraph and now reads from line 146, "All sequenced data was generated and processed using a single technical and analytical pipeline allowing for direct comparative analyses, while patient ancestries were verified using 7,472,833 markers across the genome and fastSTRUCTURE v1.0 population sub-structure analysis²², allowing for selection of patients representing **exclusively 'African'** or **'European'** genetic ancestries. We also changed **'ethnicity'** to **'genetic ancestry'** in the heading of Table 1 (see Reviewer 1 comment).

2. I am slightly confused about the numbers of individuals given in lines 117-118 (Southern

African (n = 113) and European (n = 54)), as I don't see how they match the data source recorded for the EGA repository (South Africa (n=123), Brazil (n=7) and Australia (n=53). Was the data used in this paper a combination of EGA and additional data? If this is the case, please clarify in your description.

Response: Patients were selected from the EGA source and selected for their genetic ancestries and clinical presentations. For clarification we have made the following changes to from line 113 (in blue). "From this cohort of 183 whole genomes, patient genetic ancestry was defined through population substructure analyses as 'African', predominantly Southern African, 'African-admixed' or 'European'. Using this resource, we have previously demonstrated that when interrogating for 20 genes contributing to current PCa germline testing panels, Southern African men were 2-fold less likely to present with a known pathogenic variant¹⁹, although at over 3-fold greater risk for aggressive disease presentation²⁰. Here, focusing on pathologically matched and genetically non-admixed patients, balanced for African (n = 50), specifically Southern African, and European ancestries (n = 50), largely Australian, we provided a thorough evaluation of VPPT performance for all whole genome germline predicted rare (minor allele frequency [MAF] < 1%) variants, while using this information to predict pathogenic potential for VUS."

3. Is it perhaps a bit of a stretch to describe the dataset as "multi-ethnic" as it essentially consists of two populations (Australian European and South African)? Going back to comment #2 about the "Southern African" label, I wonder whether this also implies that the data represents more diversity than they have?

Response: We have changed 'multi-ethnic' to 'African-vs-European ancestral' throughout the paper. Southern Africans is an ethnolinguistic identifier from Southern Bantu, which represents all Black South Africans (including people from Lesotho and Eswatini), Botswana yet not all Namibians although geographically Namibia falls within the southern African region. Ethnolinguistically all our study participants are Southern African or Southern Bantu. Conversely, not all South Africans are Southern African/Bantu, as they represent also South African Coloured (not included in our analyses), Indian, Malay and White South Africans. As such, our study participants who are genetically 'African' are all Southern Africans/Bantu. We do, however, understand the reviewers' comments and have reread the paper to ensure we have been as accurate as possible while ensuring simplicity in our naming.

4. Given that the cohort was selected to match patients based on pathology and ancestry, would it be possible to display the results of the population substructure methods used to remove the admixed individuals (e.g. plot from fastSTRUCTURE in the supplement)?

Response: This can be found already published in the referenced paper of Jaratlerdsiri et al., Nature 2022 in the Extended Data Fig. 1b (see below). Ancestry matching was defined as 'all African' (red) versus 'all European' (blue). See line 149 where we added the following clarification in blue; ... 'allowing for selection of patients representing exclusively 'African' or 'European' genetic ancestries.'

b, STRUCTURE analysis of bi-allelic germline variants with the logistic prior model. Model components used to explain structure in the plot are $K=5$. All spectrum of African contributions are summed and assigned as African ancestry.

5. Consider, making the difference in the mean PSA for African versus European men clearer in the sentence (line 136): Notably, PSA is excluded as an inclusion criteria as a consequence of the significant elevation in overall PSA levels at presentation in our African men (mean 448.3 versus 12.8 ng/mL [in the European men], $p = 0.001$),...

Response: The sentences from line 139 now reads as follows with addition in blue for clarity, “Notably, PSA is excluded as an inclusion criterion as a consequence of the significant elevation in overall PSA levels at presentation in our African **over European** men (mean 448.3, **22-fold greater than the NCCN European-based guidelines for aggressive disease**, versus 12.8 ng/mL, $p = 0.001$). **Elevated PSA levels** concurs with previous reports for the Southern African Prostate Cancer Study (SAPCS) for men with and also without PCa^{3,20}.”

6. I wonder whether the box containing the PV/BV identification should state “InterVar” rather than “ACMG-AMP guideline” as InterVar is more frequently used elsewhere to describe the dataset (Table 1 and main text). And perhaps also refer to them as the “merged” set to match other references in the paper (Supplementary Tables and main text)?

Response: As the guidelines are derived from ACMG-AMP and the computational tool used is InterVar, based on the reviewers comments, we have elected to use both and added ACMG-AMP throughout the text and Tables to reflect tool and guidelines, including a footnote added in Table 1.

7. Remove “for” from sentence in line 309. ...most VPPT comparison studies have excluded [for] African ancestral data⁶⁹⁻⁷¹.

Response: Removed.

8. Supplementary document includes a comment on the side from one of the authors (by Supplementary Figure 1).

Response: Removed.

Referee #3: medical genetics, field of human evolutionary genetics and genomics

Reviewer #3 (Remarks to the Author):

The authors are conducting critical analyses to evaluate and improve the performance of Variant Pathogenicity Prediction Tools (VPPTs) across ancestries, a significant step toward addressing biases and enhancing prediction accuracy in diverse populations. This work is particularly important for identifying VPPTs that perform well in African-ancestry populations, who are often underrepresented in genomic studies. VPPTs may exhibit varying levels of accuracy for different classes of genetic variants. For instance, these tools often perform well with nonsynonymous variants that directly affect protein function but may struggle with splicing or regulatory variants due to the complexity of their effects on gene expression. A focused analysis of tool performance across variant classes could identify weaknesses and guide the development of more comprehensive VPPTs. Additionally, aggressive prostate cancer (CaP) may be driven by mechanisms tied to specific variant classes. Nonsynonymous variants might indicate disruptions in key cancer-related pathways, while splicing variants could contribute to dysregulated gene expression patterns unique to aggressive disease phenotypes.

Response: We acknowledge the positive recognition for our study.

Investigating rare variant burden—a measure of the cumulative effect of rare variants within a genome, gene, or pathway—could reveal whether individuals with aggressive CaP carry a higher load of pathogenic variants compared to controls. Rare variant analyses could also identify genes or pathways enriched for rare variants, particularly in African versus European populations. Furthermore, studying the contribution of rare variants to missing heritability could clarify their role in aggressive CaP by quantifying their impact on disease heritability and explaining ancestry-specific differences. This comprehensive approach would advance the understanding of aggressive CaP genetics, highlight the need for ancestry-specific methods, and potentially inform therapeutic strategies and personalized medicine efforts.

Response: We appreciate the recognition for this importance of this research.

To further strengthen this study, the authors could explore whether common variants exhibit any differences in VPPT performance across ancestries. Common variants tend to have more robust functional annotations and are better characterized in existing genomic datasets. Investigating their predictive performance by ancestry could reveal whether the observed improvements in VPPTs are consistent across variant classes or are specific to rare or ancestry-enriched variants. This comparison could offer additional insights into the generalizability and limitations of current VPPTs and help identify opportunities for refining tools to address ancestry-specific genetic variation.

Response: As VPPT analysis is about rare variants with high-penetrance (pathogenic) rather than common variants of low penetrance, their true pathogenicity potential would be questioned and as such is not the foundational purpose ClinVar and the ACMG-AMP guidelines. We are reluctant to perform this analysis as the potential is that this is rejected by the research community. This is further highlighted by a 2020 paper that downgraded 40% of the curated common ClinVar pathogenic variants to benign, likely benign or VUS status [<https://doi.org/10.1038/s41598-019-57335-5>].

Referee #4: Genetic Medicine, population and biology

Reviewer #4 (Remarks to the Author):

The authors test the hypothesis that there exists ancestry-specific ariability in predictive power across algorithms designed for variant pathogenicity. In particular, the focus is on prostate cancer, where African individuals are less likely to have pathogenic variants but more likely to have disease. While the article is at times difficult to follow, the methodology is sound and the aim is valuable.

Response: We appreciate the recognition that our methodology is sound – we have made minor changes (in blue) to ensure it is easy to follow, especially the Table headings.

A few suggestions:

1. In table I the authors should conduct a statistical test comparing the two groups across parameters mentioned. Are they significantly different in any regard, or within the normal range of variation.

Response: Statistical significance has been reported for PSA between the groups and reads as follows from line 139, “Notably, PSA is excluded as an inclusion criterion as a consequence of the significant elevation in overall PSA levels at presentation in our African over European men (mean 448.3, 22-fold greater than the NCCN European-based guidelines for aggressive disease, versus 12.8 ng/mL, $p = 0.001$).” Additionally, it has been stated that Southern African men present on average over 6 years later, while data is matched for ISUP grading.

2. In table 3 it is not clear the ordering of algorithms, they should be organized based on type of algorithm as described in Figure 1. Also it is not clear what are the major conclusions that can be drawn from this table.

Response:

Fig. 1, the legend has been updated (see blue text) for clarity and now reads, “Fig. 1 | 35-year timeline depicting major historical events in the development of *in silico* variant pathogenicity prediction tools (VPPTs).”

Table 3, the Table heading has been updated (see blue text) for clarity and now reads, “Table 3 | Performance of 54 VPPTs for ancestry-specific (African vs European) advanced prostate cancer benchmark datasets classified by InterVar (ACMG-AMP guidelines) and ordered in descending order of sensitivity (Sen) for our African ancestral patients.”

3. While the authors do not present any code, I wonder if they implemented and can share a software pipeline that runs and compares the 54 algorithms, as this would be of great benefit to the community, in particular if it were in a stand-alone package or image that could be set up in a cloud computing environment.

Response: The core software packages used are described from line 157, “Rare variants were further annotated using ANNOVAR²³ and dbNSFP v4.7²⁴, to establish ClinVar classifications (version: 20240611)¹³, while InterVar (version: 2021-08,

<https://github.com/WGLab/InterVar>)¹² was run to establish the NCCN guideline pathogenic predictions." We have provided an appropriate reference for dbNSFP for clarification.

Only base R functions were used for the performance assessment using the formulas presented from the line 185-189. We added additional description of data analysis, including two references, from lines 190-191 which reads, "R Base Functions⁶³ and RStudio⁶⁴ (version: 2023.6.1.524) were used for data analysis and formula calculations."

4. In particular I am interested in understanding the author conclusions regarding AI-based algorithms, are they in general better or worse suited for African genome interpretation? What are the conclusions and recommendations of the authors?

Response: Artificial intelligence indeed is improving variant predicting algorithms, however dedicated processes are required from training dataset establishment to performance validation/assessment regardless of their prediction algorithms, therefore we hesitated to conclude with oversimplistic recommendations of preferred types of VPPTs, while none of the VPPTs in our workflow (Fig.2) use supervised machine learning only and concerns from the community were raised regard the issue, discussed from the line 360.

5. While it appears to be out of scope, there would be tremendous value to run the same analysis on other cancer types. Is this phenomenon unique to prostate cancer? What about multifactorial diseases and pharmacogenomics variants?

Response: While we acknowledge the additional inclusion of other cancers/diseases and similarly controls would be valuable for the community as we mentioned in this paper from line 384, the significant lack of African-focussed or regionally-matched whole genome public data makes this approach unfeasible, which further leads to difficulty in variant validation for this understudied population, including pharmacogenomics variants.

Reviewers' comments:

Reviewer #1 (Remarks to the Author):

The findings of this study will be important for effective clinical use. I have no further comments or concerns. All comments were addressed, and it is a well written paper.

Response: We appreciate the acknowledgement of the value of our work.

Reviewer #2 (Remarks to the Author):

The authors have satisfactorily responded to all of my comments and no further changes are required.

Response: We appreciate the valuable suggestions from the reviewer.

Reviewer #3 (Remarks to the Author):

This paper evaluates variant pathogenicity prediction tools (VPPTs) in the context of African genomic diversity, addressing the bias towards European populations prevalent in current genetic testing and variant interpretation. Key findings include the lower sensitivity of current VPPTs for African-derived variants, highlighting the need for ancestry-specific workflows. The study identifies VPPTs that perform better in either African or European datasets, demonstrating consistent performance across ancestries. In contrast, MutationTaster, DANN, LRT, and GERP-RS perform better in African datasets, while MutationAssessor, PROVEAN, LIST-S2, and REVEL show higher performance in European datasets. By utilizing ancestry-specific 10-VPPT workflows, the study narrowed the gap in predicting potentially deleterious variants between African and European data. Furthermore, the paper presents a method for identifying potentially oncogenic variants by combining VPPT predictions with predicted Loss of function (pLoF) and Cancer Genome Interpreter (CGI) analysis. These findings are novel and will be of significant interest to the genomics community, particularly those focused on health equity and diverse populations. The study directly addresses the underrepresentation of African populations in genomic research, a critical issue hindering the equitable application of genomic medicine. The use of a matched cohort of Southern African and European men with advanced prostate cancer strengthens the study's validity. While acknowledging the need for larger and more diverse datasets, the study provides valuable insights into VPPT performance across ancestries, emphasizing the importance of ancestral diversity in genomic studies and advocating for the development of ancestry-specific guidelines for variant interpretation. This research will contribute to achieving health equity in genomic medicine by providing practical guidance for selecting VPPTs to maximize the prediction of rare pathogenic variants.

Response: We are grateful for the reviewer's suggestions that have strengthened the manuscript. We also appreciate the acknowledgement of this work's contribution to the community.

Reviewer #4 (Remarks to the Author):

The authors have produced a significantly improved manuscript that is easy to follow, presents the problem and discovers a solution. Some minor suggestions:

1. Please number the pages in the manuscript.

Response: Added.

2. Fig 1. The earliest VPPTs predate the human genome project and were based on substitution matrices such as BLOSUM. Early manuscripts by Henikoff et al show applications in the 90's to individual gene sequences, these early methods evolved into SIFT and others. In parallel, methods based on protein structure prediction were developed, these also predate the Human genome project completion. I'd recommend updating this figure and the text describing it to reflect the early days of the field, where genes were sequenced and studied individually (without a whole genome sequence).

Response: We appreciate the acknowledgment of previous methodologies that advanced VPPT development. Text was added to Figure 1 and reads as follows, "Substitution scoring matrices and estimation of the free energy of the protein backed up subsequent VPPT development."

3. Table 3. I'm surprised by the poor reported performance of the "latest and greatest" algorithm ESM1b! Given that it is a recent and unique method, I would recommend some explanation in the manuscript results and discussion regarding this finding and what factors can be attributable for this performance of ESM1b. Based on what I know of ESM1b, the algorithm performs better on genes and gene regions where there is little MSA data available. Possibly the authors could look at variants and genes where in the ESM1b manuscript performance was reported to be higher, and confirm if this is the case for your African dataset.

Another way to sort this table and rate "performance" would be the difference in sensitivity or specificity between AF and EU populations. The main objective of this study is to identify and quantify those differences.

Response: The benchmark we used in Table 3 is not comparable to the conventional benchmark we see in other manuscripts. Given there is no public genomic resources for Africans defined in this study and ClinVar poorly represents this population (as shown in Table 1), we selected to measure the value of InterVar/ACMG-AMP guidelines, while respecting that variants from InterVar represent 'predicted' pathogenic variants. Additionally, as we applied our approach to whole genome sequenced data rather than restriction to coding variants for which VPPTs would be biased, could also compound our results. This further highlights the importance of this work as researchers advocate for the advances of whole genome sequencing over targeted gene panel approaches, [10.1186/s13073-023-01223-1](https://doi.org/10.1186/s13073-023-01223-1) for pathogenic variant discovery.

We found few studies use MCC score while most use ROC-AUC which is inflated and overoptimistic as it does not cover all four basic metrics: true positive rate, true negative rate, false positive rate and false negative rate. Furthermore, ROC-AUC is not applicable to our real-world unbalanced dataset (i.e. big difference between the number of pathogenic and benign variants) whereas MCC score is insensitive to imbalance, with further explained here: [10.1186/s13040-023-00322-4](https://doi.org/10.1186/s13040-023-00322-4).

4. The abbreviations "AF" and "EU" are non-standard, I would suggest changing them to "AFR" and "EUR" in line with other publications such as 1000 Genomes Project, GNOMAD, etc.

Response: Updated as requested.

5. Page 12, paragraph 1, line 266. I find the higher rate of VUS in European to be unexpected, can the authors elaborate on that contradictory finding? Likewise, the performance is higher for Afr variants (page 10 paragraph 1). Surprising finding, please elaborate in the discussion.

Response: Current VPPTs are still in favour European-derived variant prediction, as we shown in Supplementary Table 5: significantly more pathogenic variants are predicted for Europeans vs Africans (mean 68.1 vs 63.5, $p = 1.83E-05$). However, we still observed a 1.18x increase in rate of rare variants for Africans, as mentioned in line 268.

The slightly higher MCC score for African variants when testing with the merged benchmark dataset can be attributed to the better predicting performance on benign variants but not pathogenic variants (Supplementary Table 1).

6. Page 13, line 280. How did the authors arrive at a determination of "most commonly used" algorithms? I would suggest making that criteria clear, such as # of citations and add this to Table 2.

Response: It is based on the number of citations from a 2022 paper, as cited in line 280: "we focussed on the most commonly used VPPTs¹⁷."

7. Table 2. For clarity I would suggest adding columns to this table specifying which algorithms are ancestry-specific and other categories used in the results.

Response: As Table 2 is part of our Methods section adding results data to the Table as requested did not appear to be a natural flow for the paper. We appreciate the suggestion.